# Enhancing Virtual Real-Time Monitoring of Photovoltaic Power Systems Based on the Internet of Things

Ghedhan Boubakr [1,*], Fengshou Gu [1], Laith Farhan [2] and Andrew Ball [1]

1   Centre for Efficiency and Performance Engineering, School of Computing and Engineering, University of Huddersfield, Huddersfield HD1 3DH, UK
2   Department of Computer, College of Engineering, University of Diyala, Baquba 32001, Iraq
*   Correspondence: ghedhan.boubakr@hud.ac.uk

**Abstract:** Solar power systems have been growing globally to replace fossil fuel-based energy and reduce greenhouse gases (GHG). In addition to panel efficiency deterioration and contamination, the produced power of photovoltaic (PV) systems is intermittent due to the dependency on weather conditions, causing reliability and resiliency issues. Monitoring system parameters can help in predicting faults in time for corrective action to be taken or preventive maintenance to be applied. However, classical monitoring approaches have two main problems: neither local nor centralized monitoring support distributed PV power systems nor provide remote access capability. Therefore, this paper presents an appraisal of a remote monitoring system of PV power generation stations by utilizing the Internet of Things (IoT) and a state-of-the-art tool for virtual supervision. The proposed system allows real-time measurements of all PV system parameters, including surrounding weather conditions, which are then available at the remote control center to check and track the PV power system. The proposed technique is composed of a set of cost-effective devices and algorithms, including a PV power conditioning unit (PCU); a sensor board for measuring the variables that influence PV energy production such as irradiance and temperature, using a communication module based on Wi-Fi for data transmission; and a maximum power point tracking (MPPT) controller for enhancing the efficiency of the PV system. For validating the proposed system, different common scenarios of PV panel conditions including different shading circumstances were considered. The results show that accurate, real-time monitoring with remote access capabilities can provide timely information for predicting and diagnosing the system condition to ensure continued stable power generation and management.

**Keywords:** PV power conditioning; IoT; MPPT; smart monitoring; diagnosis system

## 1. Introduction

Increasingly, solar energy is becoming a crucial part of present and future energy policies and a vital source of energy that can be utilized and developed by humanity. However, reliability and resiliency of PV power systems are the two major concerns of most researchers attempting to eliminate the intermittent nature of energy production by solar PV because of its dependency on local weather conditions, such as irradiance and temperature. Shading effects as a result of moving objects such as clouds between the sun and the PV panels, as well as airborne dusts, bird excretions, and so on, also play a major role. Weather forecasting can help in estimating weather conditions with a high level of certainty. However, shading effects are still a problem that cannot be solved by weather prediction [1]. Real-time measurements of irradiance, ambient temperature, power, voltage, and current enable real improvements in the monitoring and controlling of PV systems for both stand-alone and grid-connected PV systems [2,3]. Real-time monitoring systems not only provide more accurate data than forecasting-based systems, they also offer an effective way to overcome system faults and help maintain system performance using

diagnostic methods for fault detection in the PV system [4]. Furthermore, these real-time monitoring systems can provide a history of operating parameters such as output power, solar radiation level, and local temperature [4]. These data can be used for later analysis and data mining for more effective PV installation and operation.

The Internet of Things (IoT) enables machine-to-machine, person-to-machine, and machine to person communication [5,6]. These state-of-the-art technologies are being realized thanks to the rapid development of various smart sensors integrated into wireless sensor networks, via miniaturization and nanotechnologies [4]. On-site, real-time measurement of PV power systems including stand-alone and grid-connected systems is an extensive and time-consuming problem for technicians responsible for the continuous monitoring and maintenance of isolated systems. Furthermore, determining power fluctuations in the generation and distribution of PV power requires considerable effort over a long period of time, and so PV power generation systems are not well-suited for remote environments. However, a survey of the current literature has shown that continuous intelligent remote monitoring of PV systems could be used to eliminate such problems [7,8].

In this paper, we introduce a virtual real-time monitoring system that utilizes IoT as one of the most cutting-edge technologies for monitoring PV power systems remotely. The proposed system diagnoses and predicts the faults of the PV panels based on data collected through real-time communication. The system combines sensors for a weather station (temperature and humidity sensor, light sensor), PCU, ESP32 platform, and MPPT for collecting data of the surrounding environment. The proposed system then transmits the collected data and tracking the peak point of the power output of the PV system, respectively. These integrated system components work collaboratively to collect and transmit the real-time measurements to be sent to wherever the monitoring stations are located.

Literature surveys of recent research contributions to monitoring of PV power systems, including comprehensive reviews of commercially available smart IoT solutions, have been carried out [4,8,9]. They have identified the applications, technologies used, and how well they performed. Based on their applications, the systems are categorized into five main categories: smart wearable, smart home, smart city, smart environment, and smart enterprise. By providing a systematic exploration of existing research and suggesting several potentially significant research directions, the authors intend to provide a guide and conceptual framework that will inspire those who are considering researching IoT applications [10–12]. Two main reasons for IoT monitoring of PV plants is the increasing expansion in power systems, especially grid-connected power systems, that makes local monitoring harder, while the advances in wire and wireless communication and Internet connection between every object in use in the daily life make local monitoring easier [13]. In the past, when there was no wireless communication, monitoring was based on local monitoring. Now, with the advanced technologies available, including smart phones and tablets that are connected all the time to the Internet, IoT has been introduced for connecting everything in one network and making the data and measurements available everywhere via the web [14]. Furthermore, the increasing rooftop installation of grid-connected PV systems needs a monitoring system which is decentralized and accessible remotely from anywhere, i.e., web-based and accessed via PCs, smart phones, and tablets. Therefore, the IoT is crucial to fulfill such needs [15]. The derived data from this study are beneficial to select a suitable solution based on the size and technical requirements of the PV monitoring project. Furthermore, IoT-based monitoring systems increase the flexibility and reduce the risk to workers and equipment as compared to the local and hardwired monitoring systems [16].

The IoT-based monitoring systems should be able to measure and collect data, transmit data to servers, and display and analyze the transmitted data [17]. Thus, reviewing IoT technologies and hardware modules recently used in monitoring PV power systems is important to select the optimum system components that best fit the current PV power system with regard to communication, speed, and accuracy. For example, a comprehensive comparison study was conducted to compare different IoT-based modules that have been

utilized in monitoring PV power systems [18]. In that study, IoT modules were classified into four main types: microcontroller-based IoT modules, such as Arduino Uno; Raspberry Pi-based modules; programmable logic controller (PLC)-based IoT; and BeagleBone-based IoT modules. The comparison included IoT communication protocols 2× I2C, 1× SPI, PCM/I2S, 2× UART, RS232, RS422, RS485, and CAN BUS. The comparison also showed the size of boards, clock speed, RAM memory and storage size, and number of GPIO pins. Moving to the IoT technologies used in monitoring PV power systems, the literature survey shows many technologies and methods that have been utilized for monitoring PV power systems based on IoT. For example, Radio Frequency Identification Technology (RFID) was used to monitor and transmit data to the cloud in a cost-effective way and with acceptable speed [19]. In addition, a simple mail transfer protocol has been used to send monitoring data to email inboxes notifying people of charges so they can take action regarding faults [20]. A typical IoT architecture can be divided into three main sections: the communication protocol by which the monitoring data can be transmitted to the server or cloud, the back-end in which the data will be stored in or fetched from the database, and the front-end that represents the user interface for showing metrics and RS displaying trends of the monitored variables [21]. Starting with the communication protocol, Wi-Fi, as one of the most widespread communication standards defined by IEEE802.11, enables connection directly to any server with different operating systems and platforms including Windows, Linux, Android, OS running on PC, or any gadget that supports Wi-Fi and also transmits data to the cloud [22]. In addition, there are many other protocols, such as ZigBee and Bluetooth, which are of low power and work for short-distance communication. In addition, there is WiMAX with the IEEE802.16 standard, which was utilized by LoRa [21]. Message Queuing Telemetry Transport (MQTT) is also one of the most common messaging protocols used in IoT applications based on TCP/IP and is a lightweight open messaging protocol that provides resource-constrained network to the users [23]. In addition, the HTTP protocol is one of the most preferred choices for exchanging data and information in IoT applications [7]. However, HTTP is not reliable when sending data to multiple recipients.

Jihua and Wang were among the first to develop a method that helps resolve managerial problems and difficulties regarding existing field maintenance practices concerning PV power generation systems [7]. These authors designed and tested a remote intelligent monitoring system using TinyOS to provide remote monitoring and forward/reverse control by a host computer, ARM gateways, and wireless sensor networks. Remote microcontroller-based monitoring of a PV power system has been developed with the capability to calculate the irradiance instead of sensing it [7]. The authors utilized the calculations to diagnose the functionality of the PV system under different operating conditions. Ranhotigamage and Mukhopadhyay were pioneers of the LoraWAN protocol to transmit real-time measurements of generated and consumed power of a PV system for determining its utilization and efficiency [24], and developments of this system are now readily available commercially from companies such as Guandong Honorinsight IoT Technology. The developers used wireless end nodes that successfully measured the power and energy at DC and AC buses for metering the power and energy production and utilization. Ansari et al. [25] carried out a review of technologies used for monitoring PV-based power systems, with a focus on data processing and transmission.

Tejwani et al. produced a design for a low-cost monitoring system that would provide information on defective PVs to enable timely repair and maintenance [26]. It was shown that monitoring the performance of a distributed system of PV panels with automated data logging using a low-cost wireless sensor network was useable on solar panel systems of up to 146 V and 15.5 A. This system could be extended to a wide range of solar cells for research and development and could provide an automatic selection of the best solutions.

An intelligent PV modular system has been developed to successfully combine control of an MPPT system with multimodal power converters using real-time wireless communication [26]. To minimize the cost, the system proposed integrating all necessary data via a single controller, which allowed a simpler supervision procedure using the smart

grid for machine-to-machine communication. Values of current and voltage for each of the PV modules in the system were periodically sampled and transmitted to the central host via a ZigBee wireless module [27]. The MPPT controller receives the data, analyzes it, and transmits relevant instructions to each PV module. The authors claim that such a PV control system benefitted from comprising only one DSP, low-cost analog controllers, and commonly found radio communication components. They also claimed that the method is suitable for monitoring and controlling the entire PV system because of the centralized control (see also [28]).

Thus, the IoT offers a promising and green solution for many modern civil and industrial application problems [29]. IoT enables machine-to-machine communication and interaction, which can enhance system management and reduce the time required for human supervision, as well as reduce the requirement of a human–machine interface [30]. The use of IoT technology can reduce the hazards and problems associated with traditional wiring when making sensor measurements and significantly improve the monitoring, performance, and efficiency of the PV system [31].

In this paper, Section 2 analyzes the findings of the literature survey, Section 3 presents the design and implementation of the proposed system, Section 4 discusses the test cases and results, and Section 5 concludes the work.

## 2. System Analysis

### 2.1. Problem Definition

The power conditioning units of remote monitoring systems of PV solar arrays are vulnerable to many challenges, such as the significant delay of implementing a repair, difficulties in maintaining a good working order, lack of flexibility, and limitations on manageability [32]. In this paper, we aim to introduce a design and implementation of an intelligent virtual monitoring system that utilizes IoT in monitoring a PV solar array. Moreover, it is capable of transmitting and storing the measured data on the cloud via a user-friendly web interface. A TCP (Transmission Control Protocol or Internet Protocol) via Wi-Fi connection was employed to connect the PV solar array and send the measured data to the cloud server (Thingspeak) in order to collect the data. This method provides system flexibility, reduces maintenance costs, and increases system efficiency [33].

### 2.2. Proposed System Features

The proposed monitoring system aims at collecting sensor readings and transmitting these measurements via Wi-Fi to the cloud database, which is accessible by a web-based user interface. In addition, the proposed unit is designed to monitor and analyze the performance of the PV systems under test using IoT devices and Virtual Instrumentation (VI) software. A solar controller circuit includes MPPT for maximizing the power output from the solar panels. The proposed design can perform early-stage fault detection in PV power systems based on IoT to reduce the time and equipment needed to effectively monitor the health of PV systems.

The overall structure and methodology of the proposed system are presented schematically in Figure 1. It uses an embedded system gateway to interface a PV solar array with the host network and transmit data from the system to a remote server via the Internet. A web page was developed to act as a graphical user interface. Data are retrieved from the cloud database and can be analyzed to determine the detailed status of the operations and functionalities of the remote PV power system. The data show the real-time status of the PV solar array, including all parameters, and can be stored to generate reports or perform further analysis [34,35]. The system's operating methodology can be divided into two independents [but internally connected elements: the hardware and the software. These two parts integrate to harvest solar power from the photovoltaic cell and transfer this energy to the battery. In addition, all metrics and sensing data, including voltage, current, ambient temperature, humidity, and irradiance, are transmitted via Wi-Fi to a cloud-based monitoring system that can be accessed remotely through a web-based user interface.

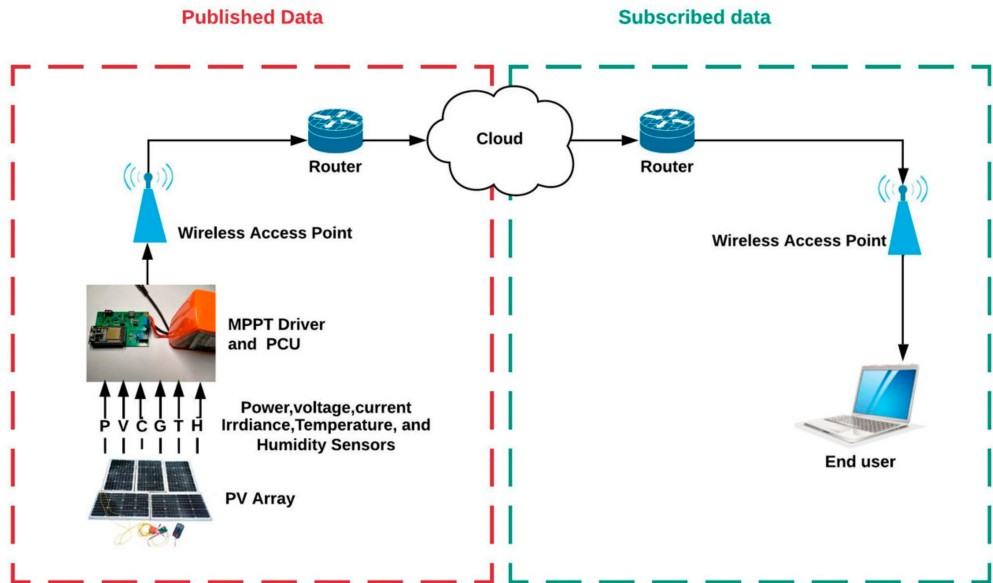

**Figure 1.** Structure of the real-time monitoring system.

### 3. System Design and Implementation

The system consists of five PV modules, sensors, and routers selected to provide a low-cost data acquisition and transmission system using a commercial microcontroller NodeMCU (ESP32), public cloud, and end-user. The sensors are connected with the ESP32 to sense the physical parameters relevant for the PV modules, as described above. So, the proposed system is transmitted the collected data to the cloud infrastructure and then to the end user through the internet.

Figure 1 depicts the schematic of the proposed IoT-based monitoring system. It shows that the structure is composed of three layers (PV arrays, system controller, and the wireless access point) that are commenced via sensors that include current, voltage, and irradiance sensors. The figure also shows the core of the monitoring system, which is the embedded gateway and the network host as the heart of the IoT system. The sensors of the PV solar array are connected to the Internet via a Wi-Fi modem through the microcontroller ESP32. The system structure includes a test PV unit, sensors, and routers to provide a low-cost DAQ and transmission system via a commercial microcontroller, public cloud, and web-based user interface. The sensors that are connected to the ESP32 can sense the operating conditions parameters of the PV power system. The microcontroller, integrated with sensors, can act as a typical IoT device that will publish the gathered data and send them to the cloud. In addition, it can disseminate data to the end-user through the cloud infrastructure. The system gateway initiates the connection to receive data from the PCU via Wi-Fi protocol. The collected data are then transferred to the cloud (Thingspeak). These data can be accessed from the cloud by the web-based user interface. This arrangement enables collecting data and uploading them to the cloud as a repository to be accessed by a web-based user interface or to be used for further analysis.

#### 3.1. Hardware Design

As mentioned earlier, the design aims to enable real-time monitoring of the PV power system with the capability of tracking the maximum power. Table 1 lists the equipment used for the proposed monitoring system.

**Table 1.** List of hardware components.

| Component | Description | Function |
|---|---|---|
| 250 W PV modules (5 × 50 W) | 5 PV modules each of 50 W capacity and 12 V monocrystalline solar panel | PV array |
| ESP32 Dev board | Main microcontroller size of 49 mm × 24.5 mm × 13 mm Connectivity type: Bluetooth and Wi-Fi | Performs logic functions and calculations |
| BQ24650 | MPPT controller 3.50 mm × 3.50 mm 12 V to 24 V automotive systems | Tracks maximum power |
| ACS712 40/30 | Current sensor 4.9 mm × 3.9 mm × 1.5 mm 66 to 185 mV/A output sensitivity | Senses PV current |
| Voltage divider circuit | Voltage-sensing board Input 18 V, output 3 V | Senses PV voltage |
| Lithium battery | Energy storage 6.4 V to 8.2 V | Backup energy storage |
| DHT11 | Temperature and humidity sensor Size 28 mm × 12 mm × 72 mm Works from 3.3 to 5 V | Senses ambient temperature and humidity |
| BH1750 | Light sensor 18.6 mm × 14.5 mm Works from 3 to 5 V | Senses irradiance incident on PV |

Figure 2 presents a single-line diagram that depicts the connections from the peripherals, including sensors for temperature, humidity, light, voltage, and current produced by the solar control system to the driver board [36,37]. The solar power input data come from five parallel solar panels, each having 20 V input voltage and 50 W power capacity. The total power is 5 × 50 = 250 W. To minimize power loss, wires of cross-sectional area 2.5 mm × 2.5 mm were used to carry the current.

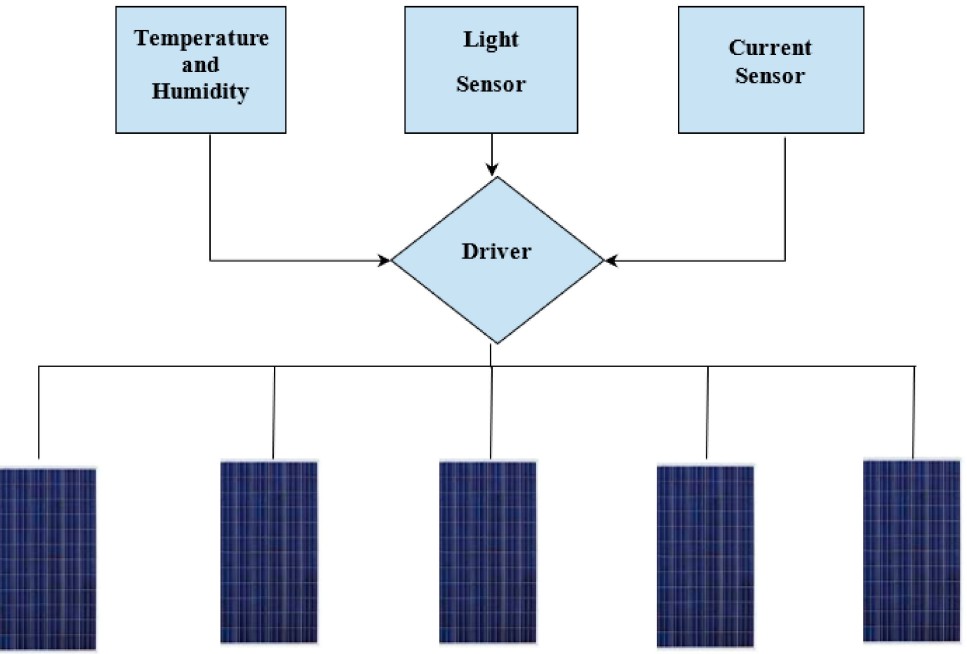

**Figure 2.** Single-line connection between devices and solar modules.

Figure 3 shows the PCB of the controller of the proposed IoT monitoring system. It shows the MPPT tracking sub-circuit and the sensor circuitry for current, voltage, humidity, and temperature.

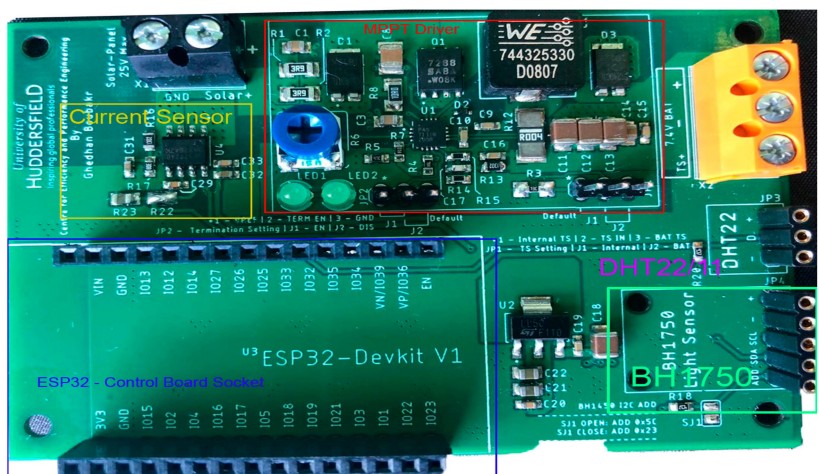

**Figure 3.** The PCB of the control circuit of the IoT monitoring system.

Figure 4 shows the PCB of the MPPT and PCU controller and related auxiliary components section. The main controller is the Texas Instrument BQ24650 charger evaluation module [38]. This was selected for the variety of features that it possesses, including determining and tracking the maximum power point, controlling the charging current, and the wide range of input voltage from 5 V to 28 V. The basis on which the BQ24650 senses the MPP is a voltage divider (Potentiometer) using the MPPSET pin, which is 1.2 V when MPPT is achieved. MPPT is based on the perturb and observe (P&O) tracking method.

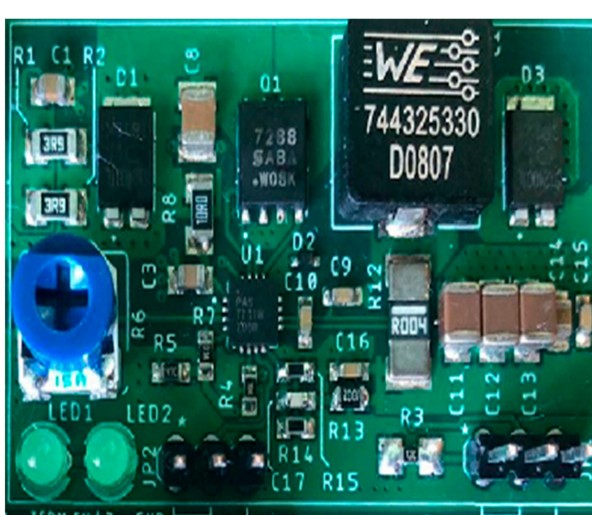

**Figure 4.** MPPT and PCU controller.

The main chip is U1 and the blue MPPT potentiometer serves to select the tracking point. The two jumpers are used to set the internal thermal sensor if the battery does not have its temperature sensor. The resistors R13, R14, and R15 are used to set the battery voltage. R13 is used as a zero-ohm resistor, but its value can be changed when a simple offset voltage is required or the exact value of R14 is not available. The formula for setting the battery voltage resistor is:

$$V_{BAT} = 2.1 \left(1 + \frac{R15}{R14}\right) \text{V} \tag{1}$$

The value 2.1 V in Equation (1) is the regulation voltage for the battery and is useful for calculating the resistor values. It is a constant value.

As discussed above, the MPPT section is set via a potentiometer which divides the voltage.

A potentiometer has three legs. Two of them are the total resistance of the potentiometer and the third one is the slider. The value of each side of the slider is the sum of the total resistance of the potentiometer.

$$MPPT_{SET} = 1.2 \left(1 + \frac{Resistance\ of\ one\ arm\ of\ the\ potentiometer}{Resistance\ of\ the\ other\ arm\ of\ the\ potentiometer}\right)V \qquad (2)$$

If the solar panel or other input source cannot provide the total power of the system and BQ24650 charger, the input voltage drops. When the voltage sensed on the MPPSET pin drops below 1.2 V, the charger maintains the input voltage by reducing the charge current.

As with $V_{BAT}$, the $MPPT_{SET}$ voltage is set using a 1.2 V regulation voltage for the MPPT. This is important for providing analog information to the driver to accommodate MPPT.

### 3.2. Implementation Setup

Figure 5 illustrates the test rig composed of five PV modules with sensors and routers to provide a low-cost DAQ and transmission system via a commercial microcontroller ESP32, public cloud, and end-user web-based user interface.

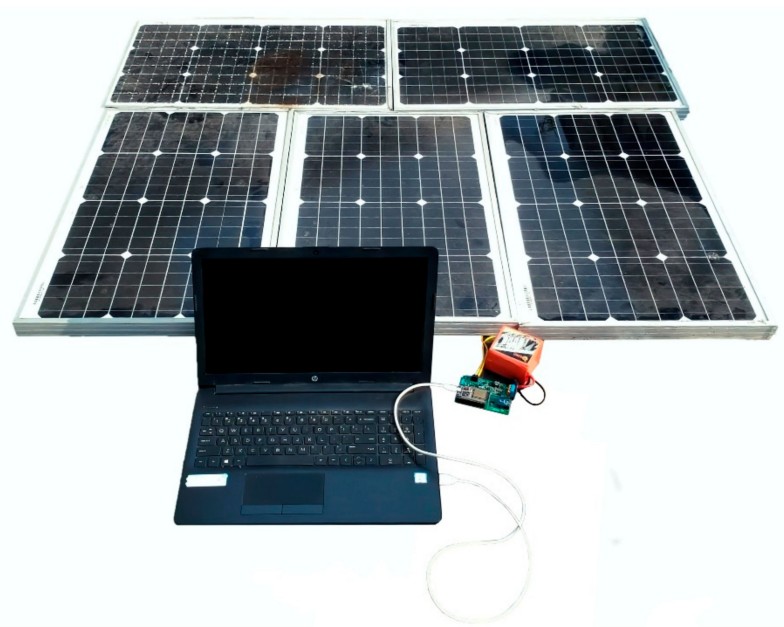

**Figure 5.** The real-time IoT-monitored PV power system.

## 4. Output Results

For validating the performance of the proposed system, five different abnormal and the normal operating conditions were tested to determine that the system recognized a faulty case and quantified the problem. In each case, the measured sensors values were transmitted via the serial communication. The collected data can be accessed by the web-based user interface or analyzed to check for faults.

### 4.1. First Case: Normal Operation

In this test case, the five PV modules were active in a no-fault condition (see Figure 6). The test considered scenarios of changing weather conditions, e.g., temperature, solar irradiance, and humidity.

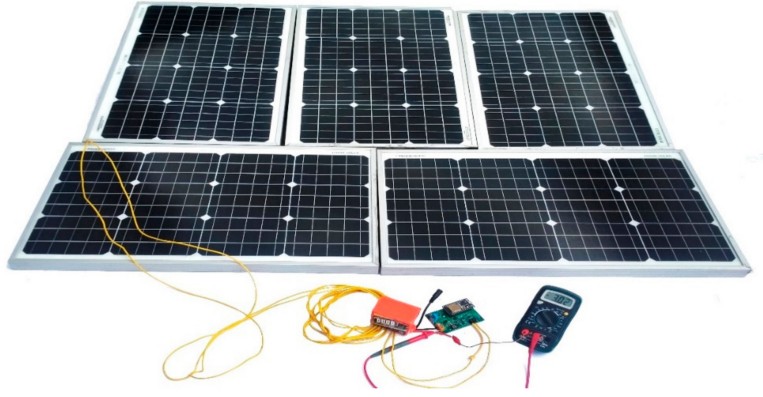

**Figure 6.** Normal operation of five active PV modules.

Figure 7 shows a typical performance of the harvesting system. We see measured values of voltage, current, power, temperature, irradiance, and humidity with time. The test period was 45 min from 4:00 p.m. to 4:45 p.m., and at 4:00 p.m., the power harvested was approximately 230 W, corresponding to a solar irradiance of 320 W/m$^2$. However, the solar irradiance dropped to 150 W/m$^2$ at 4:20 p.m., with a corresponding drop in temperature and increase in humidity. We also see that the electrical voltage output decreased significantly, while the current showed an increase thanks to the MPPT controller, which maintained the output at the maximum power point.

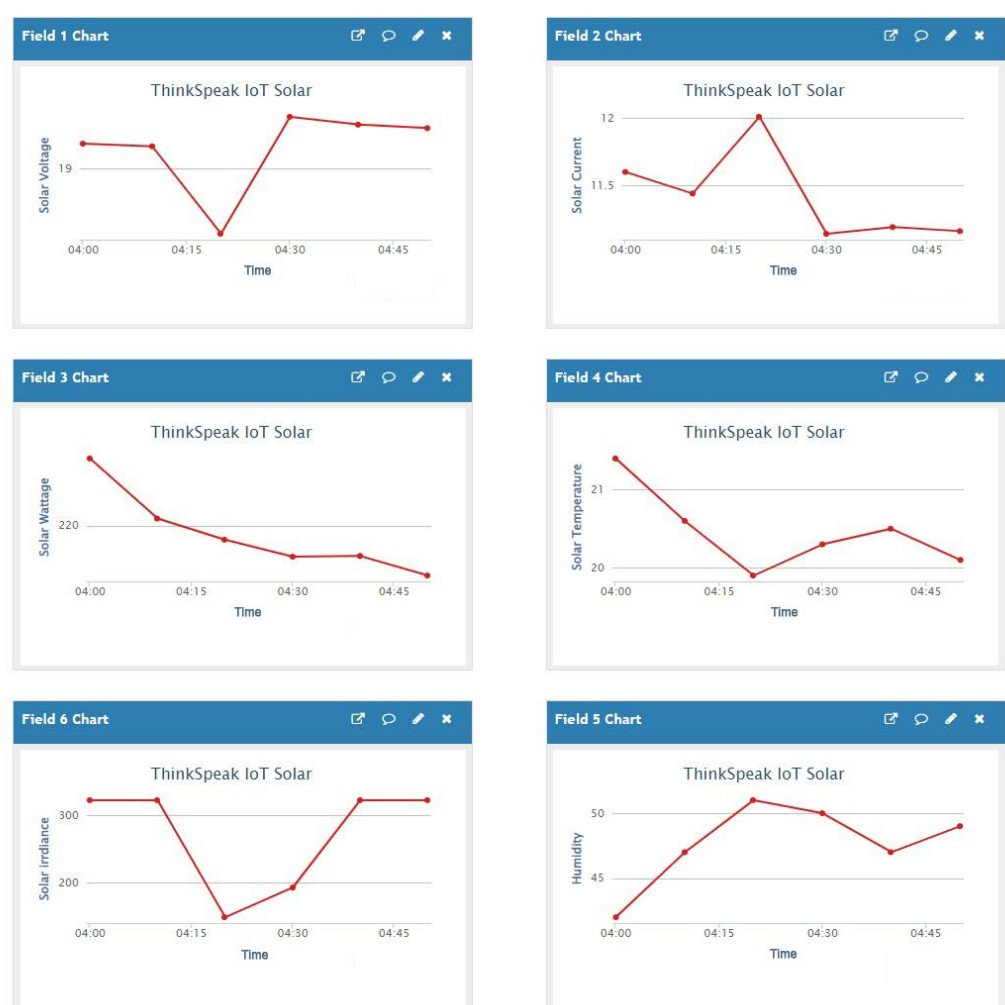

**Figure 7.** Real-time results for five active modules.

### 4.2. Second Case: Faulty Operation with Four Modules Shaded

In this test case, only one of the PV modules was active and the other four PV modules were fully shaded, as shown in Figure 8. We see the power output was 20 W at an irradiance of 200 W/m$^2$, as seen in Figure 9, due to efficient solar harvesting. In addition, it is noticeable that even in such extreme faulty conditions, the MPPT controller is still working and tracking the peak available power.

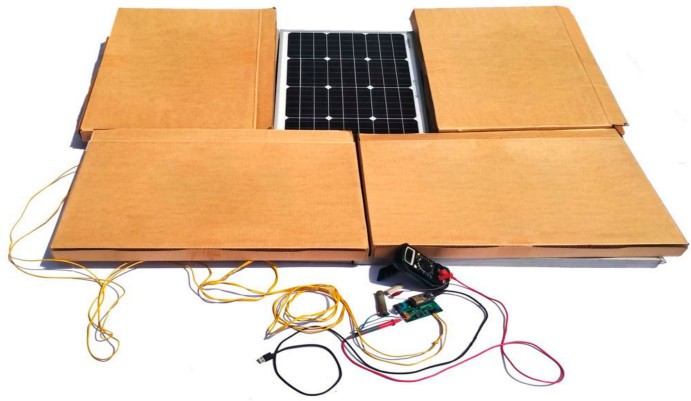

**Figure 8.** Four shaded PV modules and one active.

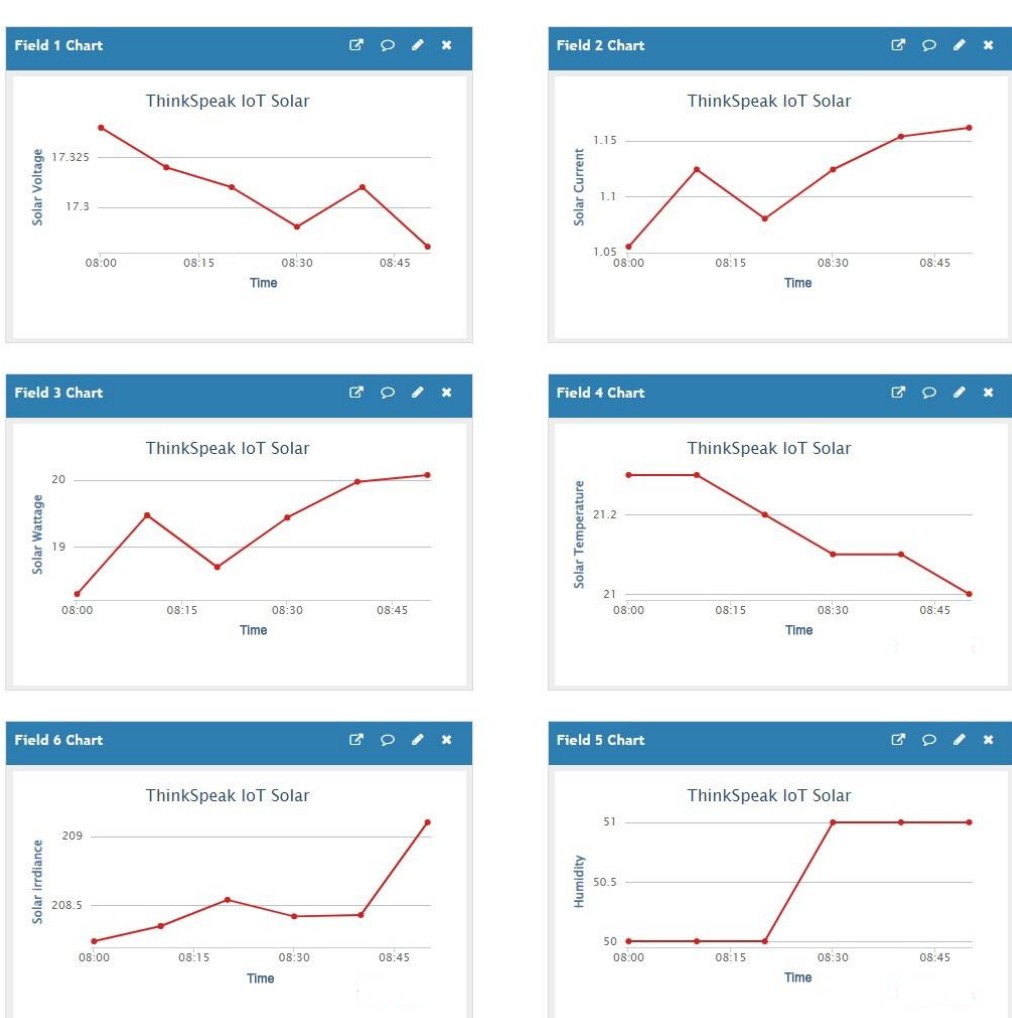

**Figure 9.** Real-time results for one active PV module.

### 4.3. Third Case: Faulty Operation, Three Modules Shaded

In this test case, there were two active PV modules and three PV modules were fully shaded, as shown in Figure 10. With only two active PV modules, the system produced a maximum of more than 70 W power output. Over the 45 min harvesting period from 7:00 a.m. to 7:45 a.m., the irradiance achieved high levels and more than 70 W of power output was achieved (see Figure 11). Again, the embedded MPPT was able to track maximum power even in the faulty conditions.

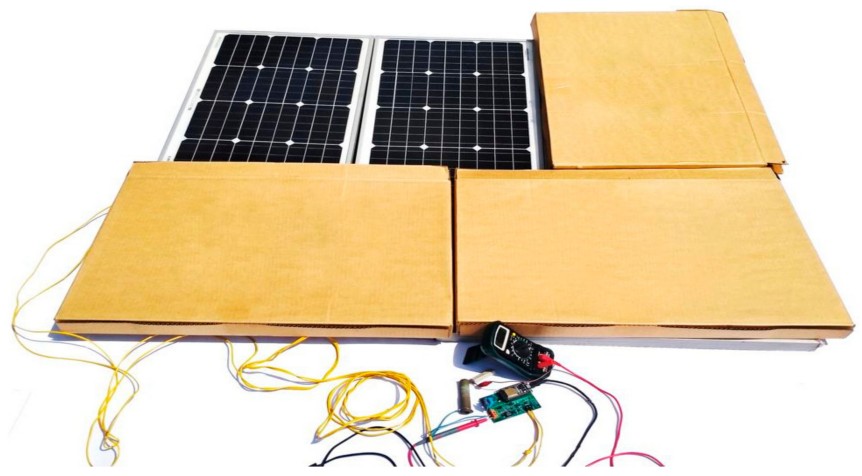

**Figure 10.** Two active PV modules.

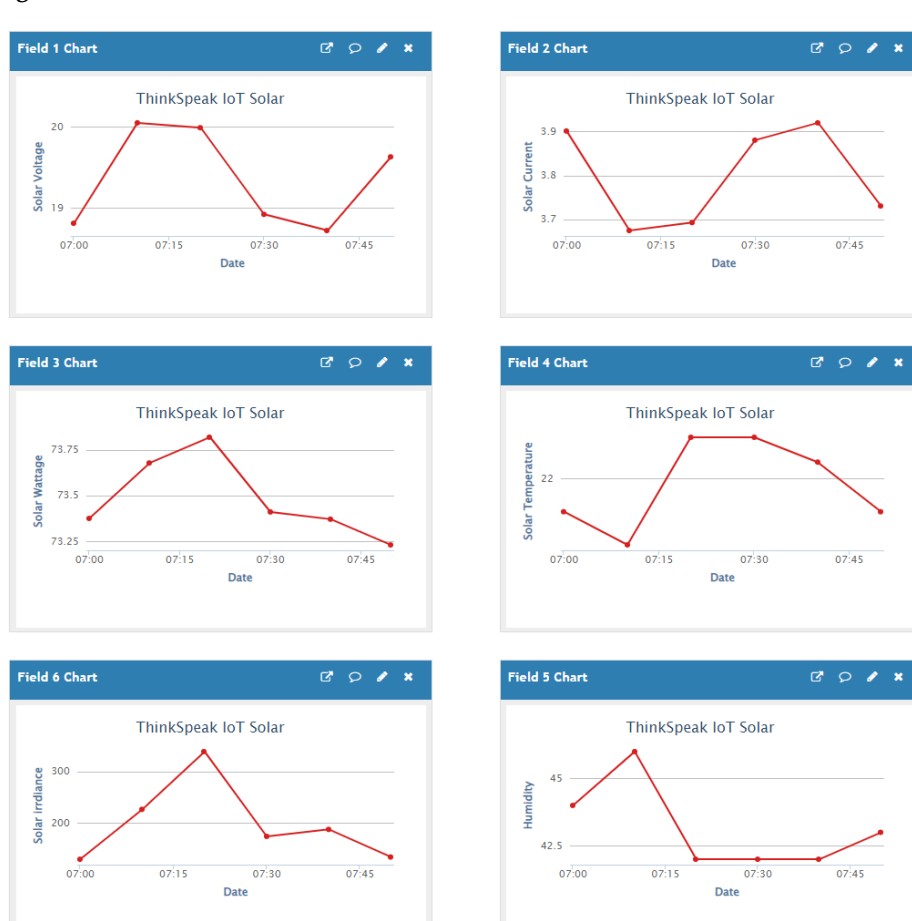

**Figure 11.** Real-time results for two active PV modules.

### 4.4. Fourth Case: Faulty Operation, Two Modules Shaded

In this test case, there were three active PV modules and two fully shaded or covered PV modules, as shown in Figure 12.

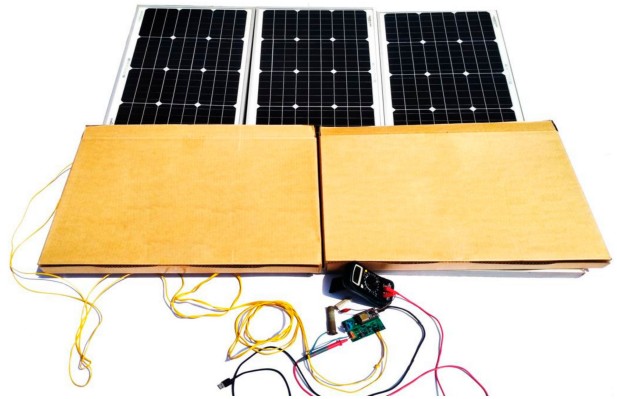

**Figure 12.** Three active PV modules with two fully shaded.

Over the 45 min test period from 6:00 p.m. to 6:45 p.m., the irradiance achieved high levels, and more than 70 W of power output was achieved. Figure 13 shows that the power output rose to a maximum value of just more than 126 W. Again, the embedded MPPT was able to track maximum power with changing temperature and solar irradiance.

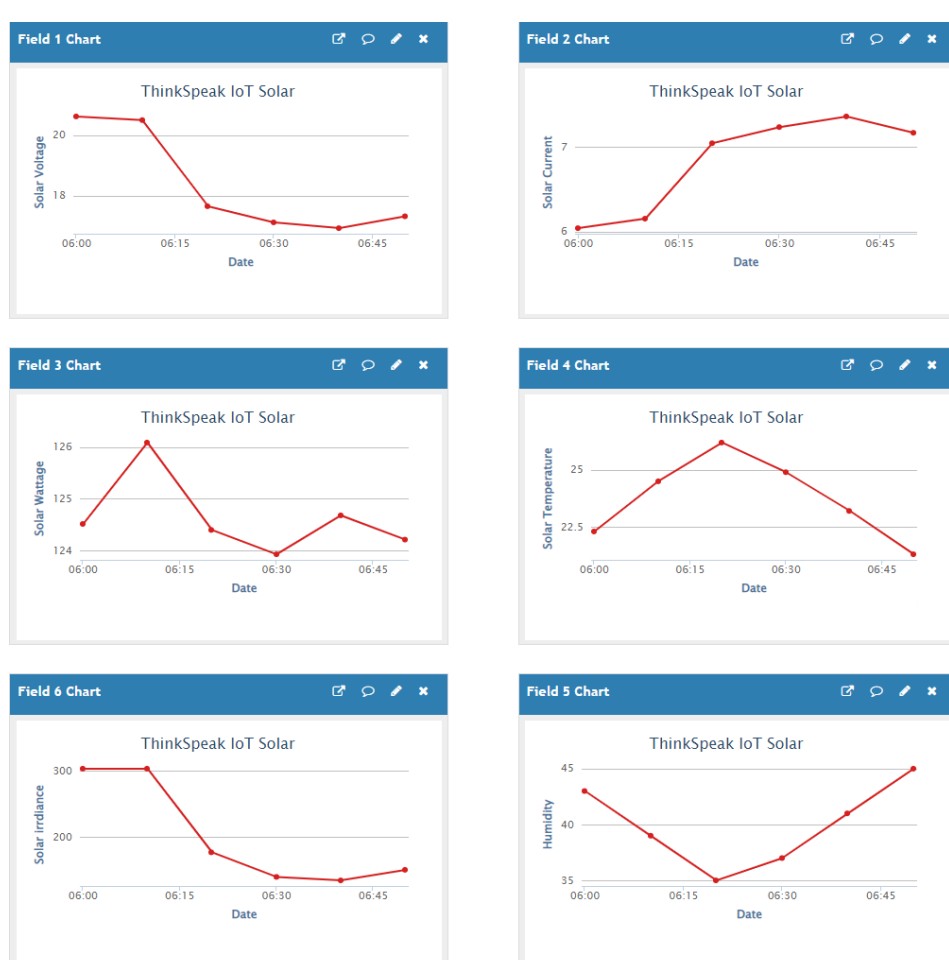

**Figure 13.** Real-time results for three active and two shaded PV modules.

### 4.5. Fifth Case: Faulty Operation, One Module Shaded

In this test case, four PV modules were active, and one PV module was shaded (see Figure 14). Output power rose to over 170 W, as shown in Figure 15. Furthermore, a drop in temperature at around 5:30 p.m. was accompanied by a decrease in current and an increase in solar voltage. Once again, the embedded MPPT was able to track the maximum power.

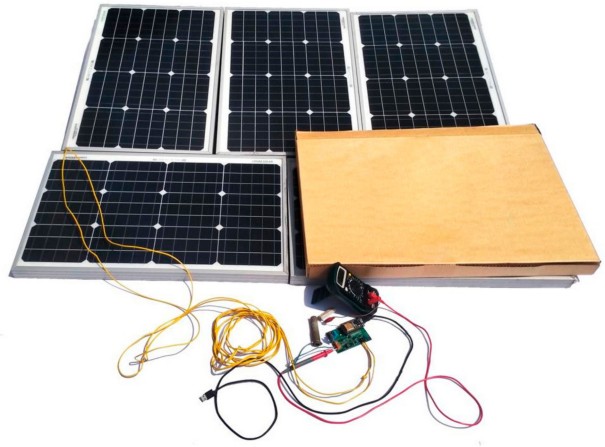

**Figure 14.** Four PV modules are active, one is shaded.

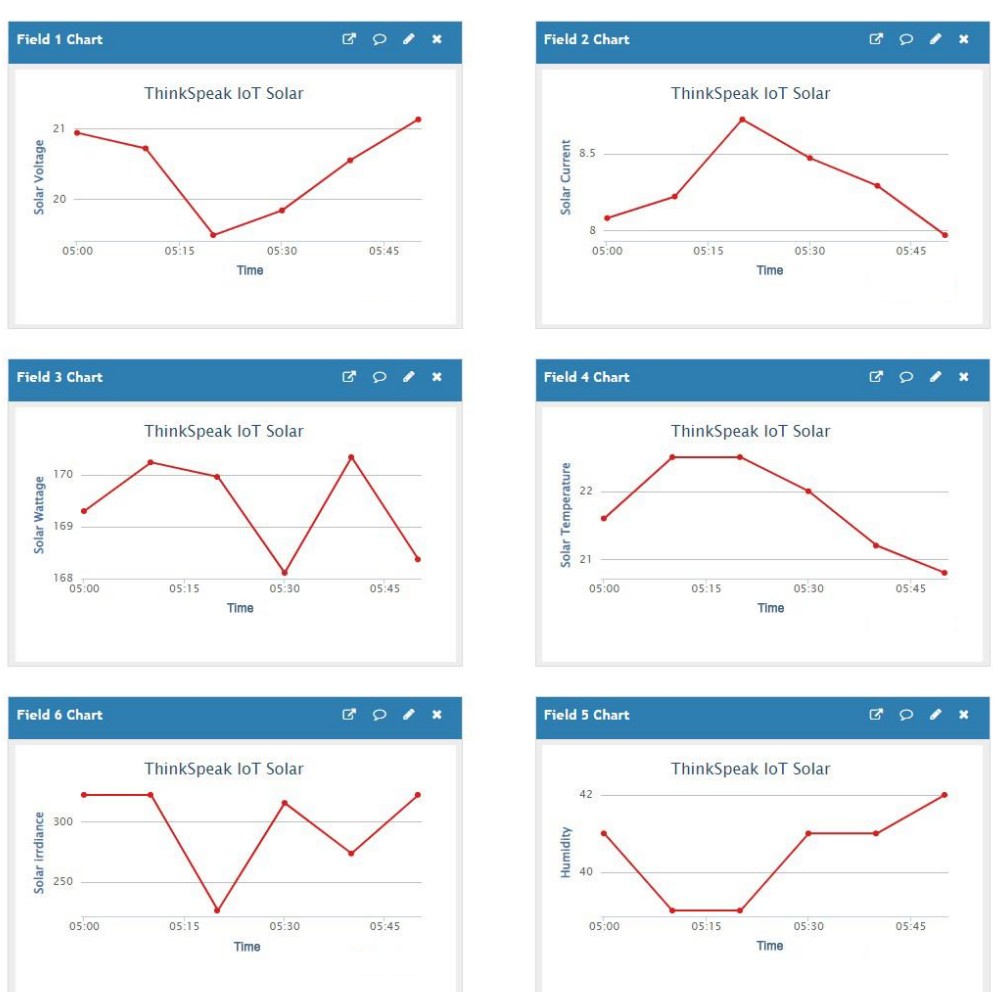

**Figure 15.** Real-time results for four active PV modules and one shaded.

## 5. Conclusions

In this paper, we introduced an IoT-based intelligent real-time monitoring system for PV power systems, which includes an MPPT controller with the goal of enhancing reliability, resilience, and system efficiency. A comprehensive review of the literature was conducted to determine the optimal components of typical IoT-based monitoring systems, including the communication protocol, back-end, and front-end for complete architecture. A complete system circuitry was utilized for monitoring the real-time system parameters, including temperature, irradiance, current, and voltage sensors. Therefore, the system is capable of accurately collecting the real-time measurements of all PV power system parameters as well as surrounding weather conditions and sending them to the cloud based on a high-speed Wi-Fi connection. The data were logged and stored in a remote server and are remotely accessible by using a web-based user interface, which has the potential to significantly reduce the time required for technicians to carry out system supervision and assist in the scheduling of plant maintenance and management. The proposed system is integrated with an embedded MPPT based on the perturb and observe (P&O) tracking method, which was able to track maximum power even under faulty conditions. Five test cases with four different fault scenarios were tested to validate the performance of the proposed system. These test cases were selected to cover the most common fault conditions related to partial and full shading. The test results show that the differences between datasets and changes in each variable show sufficient information about power generation performance and possible faulty panels, demonstrating the successful monitoring of PV power systems. In addition, in all fault cases, the fault can be recognized clearly so that corrective actions can be determined accurately. Furthermore, the MPPT controller that was included as an add-in was able to track the peak power even in faulty conditions. Our proposed IoT-based PV monitoring system integrated with the MPPT tracking method was able to immediately deliver the status of all system parameters to the remote server and the trends of real-time metrics of all system parameters in normal and faulty conditions, helping to develop predictive and preventive maintenance of PV power systems to enhance their reliability and efficiency.

**Author Contributions:** G.B.: experiment, investigation and original draft preparation; F.G., L.F. and A.B.: writing assistance. All authors have read and agreed to the published version of the manuscript.

**Funding:** This research received no external funding.

**Informed Consent Statement:** Informed consent was obtained from all subjects involved in the study.

**Conflicts of Interest:** The authors declare no conflict of interest.

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
