# Peer review of "Enhancing Virtual Real-Time Monitoring of Photovoltaic Power Systems Based on the Internet of Things"

_electronics, doi:10.3390/electronics11152469_

Round 1

Reviewer 1 Report

1-Please improve the quality of the figures 11 and 13

2-Explain the detailed principle of the proposed method

3-add more references

4-The author did not give the type of MPPT algorithm used, You must be give it and explain the principle ?and why you chose this algorithm?

5- How much sampling frequency to experimentally validate this work

Author Response

Dear Reviewer,

Please see the attachment cover letter_Reviewers.

Reviewer 2 Report

The paper was submitted by Boubakr and others in Electronics. This paper presents an appraisal of a remote monitoring station of a PV power system that utilizes the internet of things (IoT), a state-of-the-art tool for virtual supervision. The proposed technique aims at making real-time measurements of all PV system parameters including surrounding weather conditions accessible and available at the remote-control stations where corrective actions can be taken manually or automatically. The obtained results have proved accurate real-time monitoring with remote access capabilities, and diagnose and predict the performance of the system for ensuring stable power generation, as well as the storage of measurement data, is used for further analysis.

Though the idea presented in the paper is interesting however it has many flaws. I have the following comments that need to be answered sequentially.

What type of IoTs integrated with photovoltaics has been used? Please include related previous studies in the introduction. This section requires more emphasis on the importance of PV-employed IoTs. I suggest combining sections 1 and 2 concisely.

Abstract looks like a summary. Please make it brief and concise.

What type of PV system was employed for the proposed study? Please include a further explanation. Also, the description of components given in Table 1 is not complete. It should include further details.

How did the authors set the voltage to 1.2 V for the MPPT? Equations 1 and 2 are confusing. Please elaborate further.

I believe Figure 5 can be presented technically. In the current form, it is not a justifiable figure to display in a reputed journal. Also, in figure 6 are the PV modules real? Please present it in a better way. The same comment applies to Figures 7 through 15.

The conclusion does not contain any quantifiable facts. Please consider revising it completely.

Please eradicate the typos and grammatical mistakes that are consistent throughout the manuscript.

The references are not up to date and also do not follow journal guidelines. Please check carefully.

Author Response

Dear reviewer,

Please see the attachment cover lettwer_reviewer 

Regards

Round 2

Reviewer 2 Report

The manuscript has been substantially improved. I am satisfied with the author's revision. So, you can proceed with the formal acceptance.

Thank you